# Regularizing activations in neural networks via distribution matching with the Wasserstein metric

**Taejong Joo**
ESTsoft
Republic of Korea
tjoo@estsoft.com

**Donggu Kang**
ESTsoft
Republic of Korea
emppunity@gmail.com

**Byunghoon Kim**
Hanyang University
Republic of Korea
byungkim@hanyang.ac.kr

## Abstract

Regularization and normalization have become indispensable components in training deep neural networks, resulting in faster training and improved generalization performance. We propose the projected error function regularization loss (PER) that encourages activations to follow the standard normal distribution. PER randomly projects activations onto one-dimensional space and computes the regularization loss in the projected space. PER is similar to the Pseudo-Huber loss in the projected space, thus taking advantage of both $L^1$ and $L^2$ regularization losses. Besides, PER can capture the interaction between hidden units by projection vector drawn from a unit sphere. By doing so, PER minimizes the upper bound of the Wasserstein distance of order one between an empirical distribution of activations and the standard normal distribution. To the best of the authors' knowledge, this is the first work to regularize activations via distribution matching in the probability distribution space. We evaluate the proposed method on the image classification task and the word-level language modeling task.

## 1 Introduction

Training of deep neural networks is very challenging due to the vanishing and exploding gradient problem (Hochreiter, 1998; Glorot & Bengio, 2010), the presence of many flat regions and saddle points (Shalev-Shwartz et al., 2017), and the shattered gradient problem (Balduzzi et al., 2017). To remedy these issues, various methods for controlling hidden activations have been proposed such as normalization (Ioffe & Szegedy, 2015; Huang et al., 2018), regularization (Littwin & Wolf, 2018), initialization (Mishkin & Matas, 2016; Zhang et al., 2019), and architecture design (He et al., 2016).

Among various techniques of controlling activations, one well-known and successful path is controlling their first and second moments. Back in the 1990s, it has been known that the neural network training can be benefited from normalizing input statistics so that samples have zero mean and identity covariance matrix (LeCun et al., 1998; Schraudolph, 1998). This idea motivated batch normalization (BN) that considers hidden activations as the input to the next layer and normalizes scale and shift of the activations (Ioffe & Szegedy, 2015).

Recent works show the effectiveness of different sample statistics of activations for normalization and regularization. Deecke et al. (2019) and Kalayeh & Shah (2019) normalize activations to several modes with different scales and translations. Variance constancy loss (VCL) implicitly normalizes the fourth moment by minimizing the variance of sample variances, which enables adaptive mode separation or collapse based on their prior probabilities (Littwin & Wolf, 2018). BN is also extended to whiten activations (Huang et al., 2018; 2019), and to normalize general order of central moment in the sense of $L^p$ norm including $L^0$ and $L^\infty$ (Liao et al., 2016; Hoffer et al., 2018).

In this paper, we propose a projected error function regularization (PER) that regularizes activations in the Wasserstein probability distribution space. Specifically, PER pushes the distribution of activations to be close to the standard normal distribution. PER shares a similar strategy with previous approaches that dictates the ideal distribution of activations. Previous approaches, however, deal with single or few sample statistics of activations. On the contrary, PER regularizes the activations

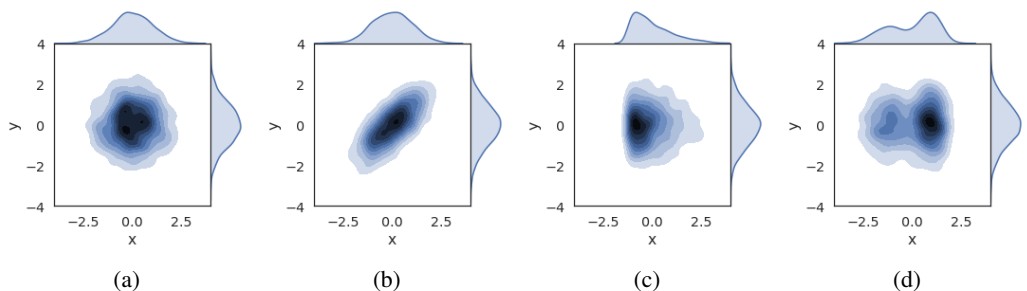

Figure 1: Limitation of statistics in terms of representing the probability distribution. In all subplots, $x$ has zero mean and unit variance and $y \sim \mathcal{N}(0,1)$. In (a) $(x,y) \sim \mathcal{N}(0, \boldsymbol{I})$. In (b), $x \sim \mathcal{N}(0,1)$ but correlated with $y$. In (c), $x$ follows a skewed distribution. In (d), $x$ follows a bi-modal distribution. Standardization cannot differentiate (a)-(d) and whitening cannot differentiate (a), (c), and (d).

by matching the probability distributions, which considers different statistics simultaneously, e.g., all orders of moments and correlation between hidden units. The extensive experiments on multiple challenging tasks show the effectiveness of PER.

## 2 RELATED WORKS

Many modern deep learning architectures employ BN as an essential building block for better performance and stable training even though its theoretical aspects of regularization and optimization are still actively investigated (Santurkar et al., 2018; Kohler et al., 2018; Bjorck et al., 2018; Yang et al., 2019). Several studies have applied the idea of BN that normalizes activations via the sample mean and the sample variance to a wide range of domains such as recurrent neural network (Lei Ba et al., 2016) and small batch size training (Wu & He, 2018).

Huang et al. (2018; 2019) propose normalization techniques whitening the activation of each layer. This additional constraint on the statistical relationship between activations improves the generalization performance of residual networks compared to BN. Although the correlation between activations are not explicitly considered, dropout prevents activations from being activated at the same time, called co-adaptation, by randomly dropping the activations (Srivastava et al., 2014), the weights (Wan et al., 2013), and the spatially connected activations (Ghiasi et al., 2018).

Considering BN as the normalization in the $L^2$ space, several works extend BN to other spaces, i.e., other norms. Streaming normalization (Liao et al., 2016) explores the normalization of a different order of central moment with $L^p$ norm for general $p$. Similarly, Hoffer et al. (2018) explores $L^1$ and $L^{\infty}$ normalization, which enable low precision computation. Littwin & Wolf (2018) proposes a regularization loss that reduces the variance of sample variances of activation that is closely related to the fourth moment.

The idea of controlling activations via statistical characteristics of activations also has motivated initialization methods. An example includes balancing variances of each layer (Glorot & Bengio, 2010; He et al., 2015), bounding scale of activation and gradient (Mishkin & Matas, 2016; Balduzzi et al., 2017; Gehring et al., 2017; Zhang et al., 2019), and norm preserving (Saxe et al., 2013). Although the desired initial state may not be maintained during training, experimental results show that they can stabilize the learning process as well.

Recently, the Wasserstein metric has gained much popularity in a wide range of applications in deep learning with some nice properties such as being a metric in a probability distribution space without requiring common supports of two distributions. For instance, it is successfully applied to a multi-labeled classification (Frogner et al., 2015), gradient flow of policy update in reinforcement learning (Zhang et al., 2018), training of generative models (Arjovsky et al., 2017; Gulrajani et al., 2017; Kolouri et al., 2019), and capturing long term semantic structure in sequence-to-sequence language model (Chen et al., 2019).

While the statistics such as mean and (co)variance are useful summaries of a probability distribution, they cannot fully represent the underlying structure of the distribution (Fig. 1). Therefore, regular-

izing or normalizing activation to follow the target distribution via statistics can be ineffective in some cases. For instance, normalizing activations via single mean and variance such as BN and decorrelated BN (Huang et al., 2018) can be inadequate in learning multimodal distribution (Bilen & Vedaldi, 2017; Deecke et al., 2019). This limitation motivates us to investigate a more general way of regularizing the distribution of activations. Instead of controlling activations via statistics, we define the target distribution and then minimize the Wasserstein distance between the activation distribution and the target distribution.

## 3 PROJECTED ERROR FUNCTION REGULARIZATION

We consider a neural network with $L$ layers each of which has $d_l$ hidden units in layer $l$. Let $\mathcal{D} = \{(\boldsymbol{x}_i, \boldsymbol{y}_i)\}_{i=1}^n$ be $n$ training samples which are assumed to be i.i.d. samples drawn from a probability distribution $P_{\mathbf{x},\mathbf{y}}$. In this paper, we consider the optimization by stochastic gradient descent with mini-batch of $b$ samples randomly drawn from $\mathcal{D}$ at each training iteration. For $i$-th element of the samples, the neural network recursively computes:

$$\boldsymbol{h}_i^l = \phi\left(\boldsymbol{W}^l \boldsymbol{h}_i^{l-1} + \boldsymbol{b}^l\right) \tag{1}$$

where $\boldsymbol{h}_i^0 = \boldsymbol{x}_i \in \mathbb{R}^{d_0}$, $\boldsymbol{h}_i^l \in \mathbb{R}^{d_l}$ is an activation in layer $l$, and $\phi$ is an activation function. In the case of recurrent neural networks (RNNs), the recursive relationship takes the form of:

$$\boldsymbol{h}_{t_i}^l = \phi\left(\boldsymbol{W}_{rec}^l \boldsymbol{h}_{t-1_i}^l + \boldsymbol{W}_{in}^l \boldsymbol{h}_{t_i}^{l-1} + \boldsymbol{b}^l\right) \tag{2}$$

where $\boldsymbol{h}_{t_i}^l$ is an activation in layer $l$ at time $t$ and $\boldsymbol{h}_{0_i}^l$ is an initial state. Without loss of generality, we focus on activations in layer $l$ of feed-forward networks and the mini-batch of samples $\{(\boldsymbol{x}_i, \boldsymbol{y}_i)\}_{i=1}^b$. Throughout this paper, we let $f^l$ be a function made by compositions of recurrent relation in equation 1 up to layer $l$, i.e., $\boldsymbol{h}_i^l = f^l(\boldsymbol{x}_i)$, and $f_j^l$ be a $j$-th output of $f^l$.

This paper proposes a new regularization loss, called projected error function regularization (PER), that encourages activations to follow the standard normal distribution. Specifically, PER directly matches the distribution of activations to the target distribution via the Wasserstein metric. Let $\mu \in \mathcal{P}(\mathbb{R}^{d_l})$ be the Gaussian measure defined as $\mu(\mathbb{A}) = \frac{1}{2^{d_l/2}} \int_{\mathbb{A}} \exp\left(-\frac{1}{2} \parallel \boldsymbol{x} \parallel^2\right) d\boldsymbol{x}$ and $\nu_{\mathbf{h}^l} = \frac{1}{b} \sum_i \delta_{\boldsymbol{h}_i^l} \in \mathcal{P}(\mathbb{R}^{d_l})$ be the empirical measure of hidden activations where $\delta_{\boldsymbol{h}_i^l}$ is the Dirac unit mass on $\boldsymbol{h}_i^l$. Then, the Wasserstein metric of order $p$ between $\mu$ and $\nu_{\mathbf{h}^l}$ is defined by:

$$W_p(\mu, \nu_{\mathbf{h}^l}) = \left(\inf_{\pi \in \prod(\mu, \nu_{\mathbf{h}^l})} \int_{\mathbb{R}^{d_l} \times \mathbb{R}^{d_l}} d^p(\boldsymbol{x}, \boldsymbol{y}) \pi(d\boldsymbol{x}, d\boldsymbol{y})\right)^{1/p} \tag{3}$$

where $\prod(\mu, \nu_{\mathbf{h}^l})$ is the set of all joint probability measures on $\mathbb{R}^{d_l} \times \mathbb{R}^{d_l}$ having the first and the second marginals $\mu$ and $\nu_{\mathbf{h}^l}$, respectively.

Because direct computation of equation 3 is intractable, we consider the sliced Wasserstein distance (Rabin et al., 2011) approximating the Wasserstein distance by projecting the high dimensional distributions onto $\mathbb{R}$ (Fig. 2). It is proved by that the sliced Wasserstein and the Wasserstein are equivalent metrics (Santambrogio, 2015; Bonnotte, 2013). The sliced Wasserstein of order one between $\mu$ and $\nu_{\mathbf{h}^l}$ can be formulated as:

$$SW_1(\mu, \nu_{\mathbf{h}^l}) = \int_{\mathbb{S}^{d-1}} W_1(\mu_{\boldsymbol{\theta}}, \nu_{\mathbf{h}_{\boldsymbol{\theta}}^l}) d\lambda(\boldsymbol{\theta}) = \int_{\mathbb{S}^{d-1}} \int_{-\infty}^{\infty} \left| F_{\mu_{\boldsymbol{\theta}}}(x) - \frac{1}{b} \sum_{i=1}^b 1_{\langle \boldsymbol{h}_i^l, \boldsymbol{\theta} \rangle \leq x} \right| dx d\lambda(\boldsymbol{\theta}) \tag{4}$$

where $\mathbb{S}^{d_l-1}$ is a unit sphere in $\mathbb{R}^{d_l}$, $\mu_{\boldsymbol{\theta}}$ and $\nu_{\mathbf{h}_{\boldsymbol{\theta}}^l}$ represent the measures projected to the angle $\boldsymbol{\theta}$, $\lambda$ is a uniform measure on $\mathbb{S}^{d-1}$, and $F_{\mu_{\boldsymbol{\theta}}}(x)$ is a cumulative distribution function of $\mu_{\boldsymbol{\theta}}$. Herein, equation 4 can be evaluated through sorting $\left\{\langle \boldsymbol{h}_i^l, \boldsymbol{\theta} \rangle\right\}_i$ for each angle $\boldsymbol{\theta}$.

While we can directly use the sliced Wasserstein in equation 4 as a regularization loss, it has a computational dependency on the batch dimension due to the sorting. The computational dependency between samples may not be desirable in distributed and large-batch training that is becoming more and more prevalent in recent years. For this reason, we remove the dependency by applying the

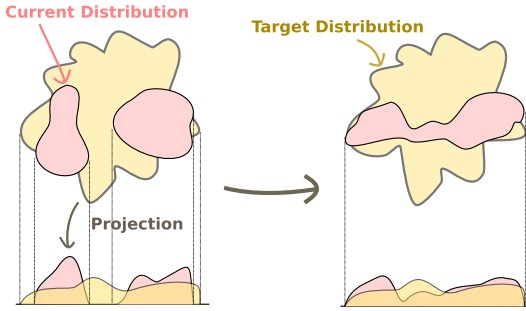

Figure 2: Illustration of minimization of the sliced Wasserstein distance between the current distribution and the target distribution. Note that it only concerns a distance in the projected dimension.

---

**Algorithm 1** Backward pass under PER

---

    **Input** The number of Monte Carlo evaluations $s$, an activation for $i$-th sample $\boldsymbol{h}_i$, the gradient of the loss $\nabla_{\boldsymbol{h}_i}\mathcal{L}$, a regularization coefficient $\lambda$

1:  $\boldsymbol{g} \leftarrow \boldsymbol{0}$
2: **for** $k \leftarrow 1$ to $s$ **do**
3:     Sample $\boldsymbol{v} \sim \mathcal{N}(\boldsymbol{0}, \boldsymbol{I})$
4:     $\boldsymbol{\theta} \leftarrow \boldsymbol{v} / \parallel \boldsymbol{v} \parallel_2$
5:     Project $h_i' \leftarrow \langle \boldsymbol{h}_i, \boldsymbol{\theta} \rangle$
6:     $g_k \leftarrow \mathrm{erf}\left(h_i'/\sqrt{2}\right)$
7:     $\boldsymbol{g} \leftarrow \boldsymbol{g} + g_k\boldsymbol{\theta}/s$
8: **end for**
9: **return** $\nabla_{\boldsymbol{h}_i}\mathcal{L} + \lambda\boldsymbol{g}$

---

Minkowski inequality to equation 4, and obtain the regularization loss $\mathcal{L}_{per}(\nu_{\mathbf{h}^l})$:

$$SW_1(\mu, \nu_{\mathbf{h}^l}) \leq \int_{\mathbb{S}^{d-1}} \int_{-\infty}^{\infty} \frac{1}{b} \sum_{i=1}^{b} \left| F_{\mu_\theta}(x) - 1_{\langle \boldsymbol{h}_i^l, \boldsymbol{\theta} \rangle \leq x} \right| dx d\lambda(\boldsymbol{\theta})$$

$$= \frac{1}{b} \sum_{i=1}^{b} \int_{\mathbb{S}^{d-1}} \left( \langle \boldsymbol{h}_i^l, \boldsymbol{\theta} \rangle \mathrm{erf}\left( \frac{\langle \boldsymbol{h}_i^l, \boldsymbol{\theta} \rangle}{\sqrt{2}} \right) + \sqrt{\frac{2}{\pi}} \exp\left( -\frac{\langle \boldsymbol{h}_i^l, \boldsymbol{\theta} \rangle^2}{2} \right) \right) d\lambda(\boldsymbol{\theta}) = \mathcal{L}_{per}(\nu_{\mathbf{h}^l}) \quad (5)$$

whose gradient with respect to $\boldsymbol{h}_i^l$ is:

$$\nabla_{\boldsymbol{h}_i^l} \mathcal{L}_{per}(\nu_{\mathbf{h}^l}) = \frac{1}{b} \mathbb{E}_{\boldsymbol{\theta} \sim U(\mathbb{S}^{d_l-1})} \left[ \mathrm{erf}\left( \langle \boldsymbol{\theta}, \boldsymbol{h}_i^l/\sqrt{2} \rangle \right) \boldsymbol{\theta} \right] \quad (6)$$

where $U(\mathbb{S}^{d_l-1})$ is the uniform distribution on $\mathbb{S}^{d_l-1}$. In this paper, expectation over $U(\mathbb{S}^{d_l-1})$ is approximated by the Monte Carlo method with $s$ number of samples. Therefore, PER results in simple modification of the backward pass as in Alg. 1.

Encouraging activations to follow the standard normal distribution can be motivated by the natural gradient (Amari, 1998). The natural gradient is the steepest descent direction in a Riemannian manifold, and it is also the direction that maximizes the probability of not increasing generalization error (Roux et al., 2008). The natural gradient is obtained by multiplying the inverse Fisher information matrix to the gradient. In Raiko et al. (2012) and Desjardins et al. (2015), under the independence assumption between forward and backward passes and activations between different layers, the Fisher information matrix is a block diagonal matrix each of which block is given by:

$$\boldsymbol{F}_l = \mathbb{E}_{(\boldsymbol{x},\boldsymbol{y}) \sim (\mathbf{x},\mathbf{y})} \left[ \frac{\partial \mathcal{L}}{\partial \mathrm{vec}(\boldsymbol{W}^l)} \frac{\partial \mathcal{L}}{\partial \mathrm{vec}(\boldsymbol{W}^l)}^T \right] = \mathbb{E}_{\boldsymbol{x}} \left[ \boldsymbol{h}^{l-1} \boldsymbol{h}^{l-1T} \right] \mathbb{E}_{(\boldsymbol{x},\boldsymbol{y})} \left[ \frac{\partial \mathcal{L}}{\partial \boldsymbol{a}^l} \frac{\partial \mathcal{L}}{\partial \boldsymbol{a}^l}^T \right] \quad (7)$$

where $\mathrm{vec}(\boldsymbol{W}^l)$ is vectorized $\boldsymbol{W}^l$, $\boldsymbol{h}^{l-1} = f^{l-1}(\boldsymbol{x})$, and $\boldsymbol{a}^l = \boldsymbol{W}^l f^{l-1}(\boldsymbol{x}) + \boldsymbol{b}^l$ for $\boldsymbol{x} \sim \mathbf{x}$.

Since computing the inverse Fisher information matrix is too expensive to perform every iterations, previous studies put efforts into developing reparametrization techniques, activation functions, and

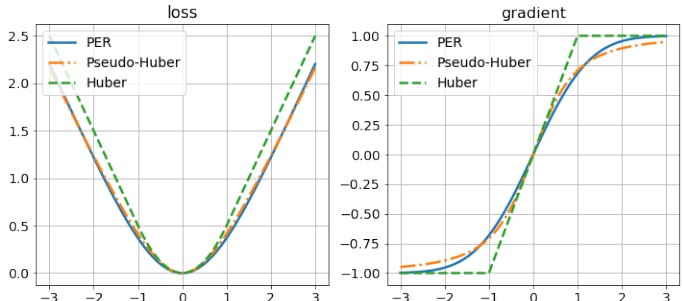

Figure 3: Illustration of PER and its gradient in $\mathbb{R}$. Herein, PER is shifted by $c$ so that $\mathcal{L}_{per}(0) - c = 0$. The Huber loss is defined as $h(x) = |x| - 0.5$ in $|x| > 1$ and $h(x) = x^2/2$ in $|x| \leq 1$ and the Pseudo-Huber loss is defined as $g(x) = \sqrt{1 + x^2} - 1$.

regularization losses to make $\boldsymbol{F}^l$ close to $\boldsymbol{I}$, thereby making the gradient close to the natural gradient. For instance, making zero mean and unit variance activations (LeCun et al., 1998; Schraudolph, 1998; Glorot & Bengio, 2010; Raiko et al., 2012; Wiesler et al., 2014) and decorrelated activations (Cogswell et al., 2016; Xiong et al., 2016; Huang et al., 2018) make $\mathbb{E}\left[\boldsymbol{h}^{l-1}\boldsymbol{h}^{l-1}{}^T\right] \approx \boldsymbol{I}$, and these techniques result in faster training and improved generalization performance. In this perspective, it is expected that PER will enjoy the same advantages by matching $\nu_{\mathbf{h}^l}$ to $\mathcal{N}(0, I)$.

## 3.1 COMPARISON TO CONTROLLING ACTIVATIONS IN $L^p$ SPACE

In this subsection, we theoretically compare PER with existing methods that control activations in $L^p$ space. $L^p(\mathbb{R}^{d_0})$ is the space of measurable functions whose $p$-th power of absolute value is Lebesgue integrable, and norm of $f \in L^p(\mathbb{R}^{d_0})$ is given by:

$$\| f \|_p = \left( \int_{\mathbb{R}^{d_0}} |f(\boldsymbol{x})|^p dP_{\mathbf{x}}(\boldsymbol{x}) \right)^{1/p} < \infty \tag{8}$$

where $P_{\mathbf{x}}$ is the unknown probability distribution generating training samples $\{\boldsymbol{x}_i\}_{i=1}^n$. Since we have no access to $P_{\mathbf{x}}$, it is approximated by the empirical measure of mini-batch samples.

The $L^p$ norm is widely used in the literature for regularization and normalization of neural networks. For instance, activation norm regularization (Merity et al., 2017a) penalizes $L^2$ norm of activations. As another example, BN and its $p$-th order generalization use $L^p$ norm such that the norm of the centralized activation, or pre-activation, is bounded:

$$\psi(h_{ij}^l) = \gamma_j^l \xi(h_{ij}^l) + \beta_j^l, \quad \xi(h_{ij}^l) = \frac{h_{ij}^l - \bar{\mu}_j}{\left( \sum_k \frac{1}{b} |h_{kj}^l - \bar{\mu}_j|^p \right)^{1/p}} \tag{9}$$

where $h_{ij}^l$ is $j$-th unit of $\boldsymbol{h}_i^l$, $\bar{\mu}_j = \frac{1}{b}\sum_k h_{kj}^l$ is the sample mean, $\beta_j^l$ is a learnable shift parameter, and $\gamma_j^l$ is a learnable scale parameters. Herein, we have $\| \xi \circ f_j^l \|_p = 1$ for any unit $j$ and any empirical measure, thus $\| \psi \|_p \leq \| \gamma_j^l \xi \circ f_j^l \|_p + \| \beta_j^l \|_p = |\gamma_j^l| + |\beta_j^l|$.

PER differs from $L^p$ norm-based approaches in two aspects. First, PER can be considered as $L^p$ norm with adaptive order in the projected space because it is very similar to the Pseudo-Huber loss in one-dimensional space (Fig. 3). Herein, the Pseudo-Huber loss is a smooth approximation of the Huber loss (Huber, 1964). Therefore, PER smoothly changes its behavior between $L^1$ and $L^2$ norms, making the regularization loss sensitive to small values and insensitive to outliers with large values. However, the previous approaches use predetermined order $p$, which makes the norm to change insensitively in the near-zero region when $p \leq 1$ or to explode in large value region when $p > 1$.

Second, PER captures the interaction between hidden units by projection vectors, unlike $L^p$ norm. To see this, let $\| f^l \|_p^p = \frac{1}{b}\sum_{i,j} |h_{ij}^l|^p = \frac{1}{b}\sum_{i,j} |\langle \boldsymbol{h}_i^l, \boldsymbol{e}_j \rangle|^p$ where $\{\boldsymbol{e}_j\}_{j=1}^{d_l}$ is the natural basis of

Table 1: Top-1 error rates of ResNets on CIFAR-10. Lower is better. All numbers are rounded to two decimal places. Boldface indicates the minimum error. * and ** are results from Zhang et al. (2019) and He et al. (2016), respectively.

| Model | Method | Test error |
|-------|--------|-----------|
| ResNet-56 | Vanilla | 7.21 |
| | BN | 6.95 |
| | PER | **6.72** |
| ResNet-110 | Vanilla | 6.90 (7.24*) |
| | BN | 6.62 (6.61**) |
| | PER | **6.19** |

Table 2: Top-1 error rates of 11-layer CNNs on tiny ImageNet. Lower is better. All numbers are rounded to two decimal places. Boldface indicates the minimum error. Numbers in parentheses represent results in Littwin & Wolf (2018).

| Method | Test error |
|--------|-----------|
| Vanilla | 37.45 (39.22) |
| BN | 39.22 (40.02) |
| VCL | (37.30) |
| PER | **36.74** |

$\mathbb{R}^{d_l}$. That is, the norm computes the regularization loss, or the normalizer, of activations with the natural basis as a projection vector. However, PER uses general projection vectors $\theta \sim U(\mathbb{S}^{d_l-1})$, capturing the interaction between hidden units when computing the regularization loss. These two differences make PER more delicate criterion for regularizing activations in deep neural networks than $L^p$ norm, as we will show in the next section.

## 4 EXPERIMENTS

This section illustrates the effectiveness of PER through experiments on different benchmark tasks with various datasets and architectures. We compare PER with BN normalizing the first and second moments and VCL regularizing the fourth moments. PER is also compared with $L^1$ and $L^2$ activation norm regularizations that behave similarly in some regions of the projected space. We then analyze the computational complexity PER and the impact of PER on the distribution of activations. Throughout all experiments, we use 256 number of slices and the same regularization coefficient for the regularization losses computed in each layer.

### 4.1 IMAGE CLASSIFICATION IN CIFAR-10, CIFAR-100, AND TINY IMAGENET

We evaluate PER in image classification task in CIFAR (Krizhevsky et al., 2009) and a subset of ImageNet (Russakovsky et al., 2015), called tiny ImageNet. We first evaluate PER with ResNet (He et al., 2016) in CIFAR-10 and compare it with BN and a vanilla network initialized by fixup initialization (Zhang et al., 2019). We match the experimental details in training under BN with He et al. (2016) and under PER and vanilla with Zhang et al. (2019), and we obtain similar performances presented in the papers. Herein, we search the regularization coefficient over { 3e-4, 1e-4, 3e-5, 1e-5 }. Table 1 presents results of CIFAR-10 experiments with ResNet-56 and ResNet-110. PER outperforms BN as well as vanilla networks in both architectures. Especially, PER improves the test errors by 0.49 % and 0.71% in ResNet-56 and ResNet-110 without BN, respectively.

We also performed experiments on an 11-layer convolutional neural network (11-layer CNN) examined in VCL (Littwin & Wolf, 2018). This architecture is originally proposed in Clevert et al. (2016). Following Littwin & Wolf (2018), we perform experiments on 11-layer CNNs with ELU, ReLU, and Leaky ReLU activations, and match experimental details in Littwin & Wolf (2018) except that we used 10x less learning rate for bias parameters and additional scalar bias after ReLU and Leaky ReLU based on Zhang et al. (2019). By doing so, we obtain similar results presented in Littwin & Wolf (2018). Again, a search space of the regularization coefficient is { 3e-4, 1e-4, 3e-5, 1e-5 }. For ReLU and Leaky ReLU in CIFAR-100, however, we additionally search { 3e-6, 1e-6, 3e-7, 1e-7 } because of divergence of training with PER in these setting. As shown in Table 3, PER shows the best performances on four out of six experiments. In other cases, PER gives compatible performances to BN or VCL, giving 0.16 % less than the best performances.

Following Littwin & Wolf (2018), PER is also evaluated on tiny ImageNet. In this experiment, the number of convolutional filters in each layer is doubled. Due to the limited time and resources, we

Table 3: Top-1 error rates of 11-layer CNNs on CIFAR-10 and CIFAR-100. Lower is better. All numbers are rounded to two decimal places. Boldface indicates the minimum error. Numbers in parentheses represent results in Littwin & Wolf (2018).

| Activation | Method | CIFAR-10 | CIFAR-100 |
|---|---|---|---|
| ReLU | Vanilla | 8.43 (8.36) | 29.45 (32.80) |
| | BN | 7.53 (7.78) | **29.13** (29.10) |
| | VCL | 7.80 (7.80) | 30.30 (30.30) |
| | PER | **7.21** | 29.29 |
| LeakyReLU | Vanilla | 6.73 (6.70) | 26.50 (26.80) |
| | BN | 6.38 (7.08) | 26.83 (27.20) |
| | VCL | 6.45 (6.45) | 26.30 (26.30) |
| | PER | **6.29** | **25.50** |
| ELU | Vanilla | 6.74 (6.98) | 27.53 (28.70) |
| | BN | 6.69 (6.63) | 26.60 (26.90) |
| | VCL | **6.26** (6.15) | 25.86 (25.60) |
| | PER | 6.42 | **25.73** |

conduct experiments only with ELU that gives good performances for PER, BN, and VCL in CIFAR. As shown in Table 2, PER is also effective in the larger model in the larger image classification dataset.

## 4.2 LANGUAGE MODELING IN PTB AND WIKITEXT2

We evaluate PER in word-level language modeling task in PTB (Mikolov et al., 2010) and WikiText2 (Merity et al., 2017b). We apply PER to LSTM with two layers having 650 hidden units with and without reuse embedding (RE) proposed in Inan et al. (2017) and Press & Wolf (2016), and variational dropout (VD) proposed in Gal & Ghahramani (2016). We used the same configurations with Merity et al. (2017a) and failed to reproduce the results in Merity et al. (2017a). Especially, when we rescale gradient when its norm exceeds 10, we observed divergence or bad performance (almost 2x perplexity compared to the published result). Therefore, we rescale gradient with norm over 0.25 instead of 10 based on the default hyperparameter of the PyTorch word-level language model[1] that is also mentioned in Merity et al. (2017a). We also train the networks for 60 epochs instead of 80 epochs since validation perplexity is not improved after 60 epochs in most cases. In this task, PER is compared with recurrent BN (RBN; Cooijmans et al., 2017) because BN is not directly applicable to LSTM. We also compare PER with $L^1$ and $L^2$ activation norm regularizations. Herein, the search space of regularization coefficients of PER, $L^1$ regularization, and $L^2$ regularization is {3e-4, 1e-4, 3e-5 }. For $L^1$ and $L^2$ penalties in PTB, we search additional coefficients over { 1e-5, 3e-6, 1e-6, 3e-6, 1e-6, 3e-7, 1e-7 } because the searched coefficients seem to constrain the capacity.

We list in Table 4 the perplexities of methods on PTB and WikiText2. While all regularization techniques show regularization effects by giving improved test perplexity, PER gives the best test perplexity except LSTM and RE-VD-LSTM in the PTB dataset wherein PER is the second-best method. We also note that naively applying RBN often reduces performance. For instance, RBN increases test perplexity of VD-LSTM by about 5 in PTB and WikiText2.

## 4.3 ANALYSIS

In this subsection, we analyze the computational complexity of PER and its impact on closeness to the standard normal distribution in the 11-layer CNN.

---

[1]Available in `https://github.com/pytorch/examples/tree/master/word_language_model`

Table 4: Validation and test perplexities on PTB and WikiText2. Lower is better. All numbers are rounded to one decimal place. Boldface indicates minimum perplexity.

| Model | Method | PTB | | WikiText2 | |
|---|---|---|---|---|---|
| | | Valid | Test | Valid | Test |
| LSTM | Vanilla | 123.2 | 122.0 | 138.9 | 132.7 |
| | $L^1$ penalty | 119.6 | **114.1** | 137.7 | 130.0 |
| | $L^2$ penalty | 120.5 | 115.2 | 136.0 | 131.1 |
| | RBN | **118.2** | 115.1 | 156.2 | 148.3 |
| | PER | 118.5 | 114.5 | **134.2** | **129.6** |
| RE-LSTM | Vanilla | 114.1 | 112.2 | 129.2 | 123.2 |
| | $L^1$ penalty | 112.2 | 108.5 | 128.6 | 122.7 |
| | $L^2$ penalty | 116.6 | **108.2** | 126.5 | 123.3 |
| | RBN | 113.6 | 110.4 | 138.1 | 131.6 |
| | PER | **110.0** | 108.5 | **123.2** | **117.4** |
| VD-LSTM | Vanilla | 84.9 | 81.1 | 99.6 | 94.5 |
| | $L^1$ penalty | 84.9 | 81.5 | 98.2 | 92.9 |
| | $L^2$ penalty | 84.5 | 81.2 | 98.8 | 94.2 |
| | RBN | 89.7 | 86.4 | 104.3 | 99.4 |
| | PER | **84.1** | **80.7** | **98.1** | **92.6** |
| RE-VD-LSTM | Vanilla | 78.9 | 75.7 | 91.4 | 86.4 |
| | $L^1$ penalty | 78.3 | 75.1 | 90.5 | 86.1 |
| | $L^2$ penalty | 79.2 | 75.8 | **90.3** | 86.1 |
| | RBN | 83.7 | 80.5 | 95.5 | 90.5 |
| | PER | **78.1** | **74.9** | 90.6 | **85.9** |

### 4.3.1 COMPUTATIONAL COMPLEXITY

PER has no additional parameters. However, BN and VCL require additional parameters for each channel and each location and channel in every layer, respectively; that is, 2.5K and 350K number of parameters are introduced in BN and VCL in the 11-layer CNN, respectively. In terms of time complexity, PER has the complexity of $O(bd_l s)$ for projection operation in each layer $l$. On the other hand, BN and VCL have $O(bd_l)$ complexities. In our benchmarking, each training iteration takes 0.071 seconds for a vanilla network, 0.083 seconds for BN, 0.087 for VCL, and 0.093 seconds for PER on a single NVIDIA TITAN X. Even though PER requires slightly more training time than BN and VCL, this disadvantage can be mitigated by computation of PER is only required in training and PER does not have additional parameters.

### 4.3.2 CLOSENESS TO THE STANDARD NORMAL DISTRIBUTION

To examine the effect of PER on the closeness to $\mathcal{N}(\mathbf{0}, \mathbf{I})$, we analyze the distribution of activations in 11-layer CNN in different perspectives. We first analyze the distribution of a single activation $h_j^l$ for some unit $j$ and layer $l$ (Fig. 4). We observe that changes in probability distributions between two consecutive epochs are small under BN because BN bound the $L^2$ norm of activations into learned parameters. On the contrary, activation distributions under vanilla and PER are jiggled between two consecutive epochs. However, PER prevents the variance explosion and pushes the mean to zero. As shown in Fig. 4, while variances of $\nu_{\boldsymbol{h}_j^6}$ under both PER and Vanilla are very high at the beginning of training, the variance keeps moving towards one under PER during training. Similarly, PER recovers biased means of $\nu_{\boldsymbol{h}_j^3}$ and $\nu_{\boldsymbol{h}_j^9}$ at the early stage of learning.

To precisely evaluate closeness to the standard normal distribution, we also analyze $SW_1(\mathcal{N}(\mathbf{0}, \mathbf{I}), \nu_{\mathbf{h}^l})$ at each epoch (Fig. 5). Herein, the sliced Wasserstein distance is computed by approximating the Gaussian measure using the empirical measure of samples drawn from $\mathcal{N}(\mathbf{0}, \mathbf{I})$ as in Rabin et al. (2011). As similar to the previous result, while BN $\beta_j^l = 0$ and $\gamma_j^l = 1$ at initial state gives small $SW_1(\mathcal{N}(\mathbf{0}, \mathbf{I}), \nu_{\mathbf{h}^l})$ in early stage of training, PER also can effectively control

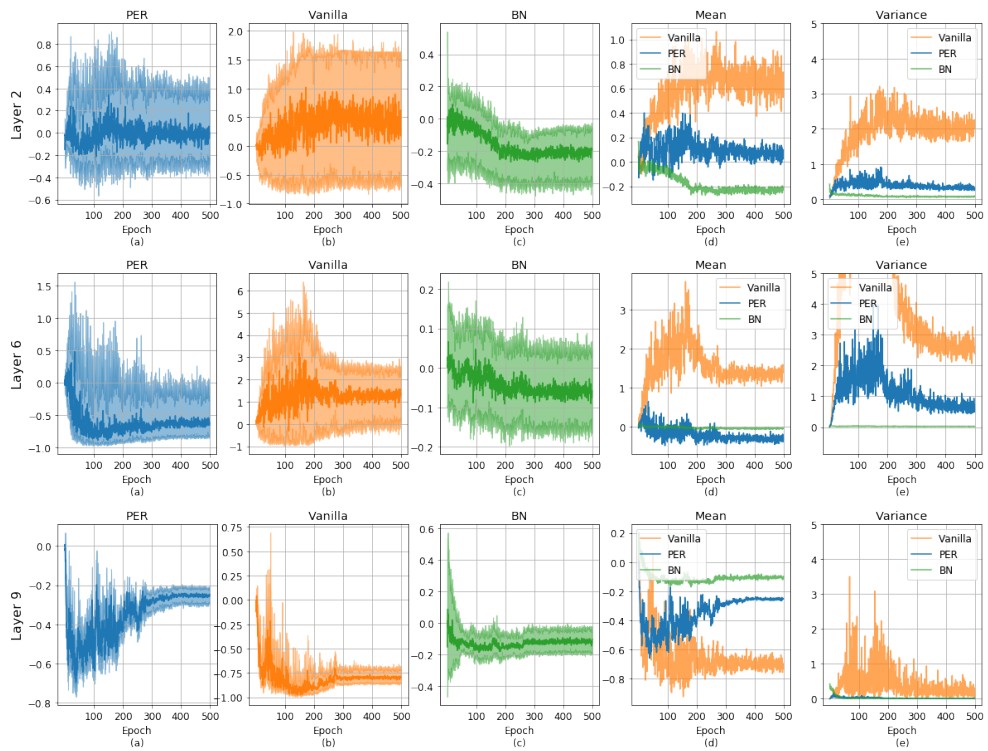

Figure 4: Evolution of distributions of $\nu_{\boldsymbol{h}_i^3}$, $\nu_{\boldsymbol{h}_j^6}$, and $\nu_{\boldsymbol{h}_j^9}$ for fixed randomly drawn $i, j, k$ on training set. (a)-(c) represent values (0.25, 0.5, 0.75) quantiles under PER, vanilla, and BN. (d) and (e) represent the sample mean and the sample variance of activations. Variance is clipped at 5 for better visualization.

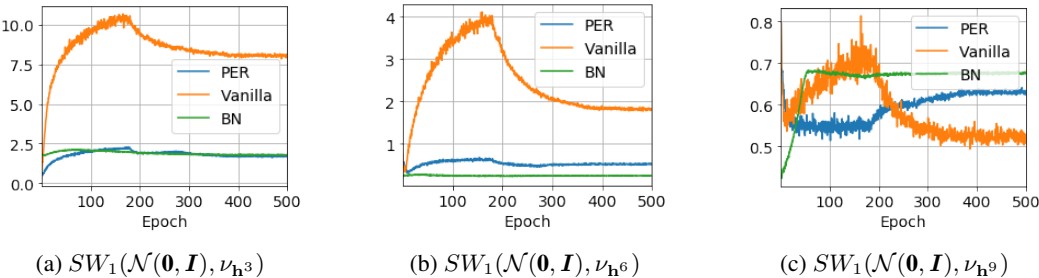

(a) $SW_1(\mathcal{N}(\boldsymbol{0}, \boldsymbol{I}), \nu_{\mathbf{h}^3})$  (b) $SW_1(\mathcal{N}(\boldsymbol{0}, \boldsymbol{I}), \nu_{\mathbf{h}^6})$  (c) $SW_1(\mathcal{N}(\boldsymbol{0}, \boldsymbol{I}), \nu_{\mathbf{h}^9})$

Figure 5: Closeness to $\mathcal{N}(0, \boldsymbol{I})$ in the Wasserstein probability distribution space.

the distribution without such normalization. This confirms that PER prevents the distribution of activation to be drifted away from the target distribution.

## 5 CONCLUSION

We proposed the regularization loss that minimizes the upper bound of the 1-Wasserstein distance between the standard normal distribution and the distribution of activations. In image classification and language modeling experiments, PER gives marginal but consistent improvements over methods based on sample statistics (BN and VCL) as well as $L^1$ and $L^2$ activation regularization methods. The analysis of changes in activations' distribution during training verifies that PER can stabilize the probability distribution of activations without normalization. Considering that the regularization loss can be easily applied to a wide range of tasks without changing architectures or training strategies

unlike BN, we believe that the results indicate the valuable potential of regularizing networks in the probability distribution space as a future direction of research.

The idea of regularizing activations with the metric in probability distribution space can be extended to many useful applications. For instance, one can utilize task-specific prior when determining a target distribution, e.g., the Laplace distribution for making sparse activation. The empirical distribution of activations computed by a pretrained network can also be used as a target distribution to prevent catastrophic forgetting. In this case, the activation distribution can be regularized so that it does not drift away from the activation distribution learned in the previous task as different from previous approaches constrains the changes in the the function $L^2$ space of logits (Benjamin et al., 2019).

ACKNOWLEDGMENTS

We would like to thank Min-Gwan Seo, Dong-Hyun Lee, Dongmin Shin, and anonymous reviewers for the discussions and suggestions.

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
