# OpenReview forum: "Regularizing activations in neural networks via distribution matching with the Wasserstein metric"
_ICLR.cc/2020/Conference — Accept (Poster)_

### Official Review · AnonReviewer3 · 2019-10-23
**Official Blind Review #3**

**Rating:** 3

**Review:**

This paper proposes a new method to normalize activations in neural networks, based on an upper bound of the sliced Wasserstein distance between the empirical activation distribution and a standard Gaussian distribution. I think this feels like a "borderline" case to me. The paper clearly has merits, at the same time there're some issues to be addressed.

Pros:
- The idea is clearly presented.
- Better performance than BN is achieved in many experiments.
- Empirical evidence in Section 4.3 looks good, suggesting the proposed method does do the job as expected. The means and variances stabilize as training progresses.

Cons:
- While the method based on sliced Wasserstein distances sounds new, the novelty seems limited since the idea of whitening the activation distribution to unit Gaussian was introduced before as mentioned by the authors. The paper claims the random projection may capture “interaction between hidden units”, but it seems the method proposed in e.g. Huang et. al. 2018 also has projection matrices that might be doing similar things?

- I’m concerned about the actual computation cost of the proposed method. Although the method does not introduce any additional parameter compared to BN or VCL, it seems to require multiple random projections for each layer (s=256 in the experiments)? This could be much slower than the BN. A clarification/comparison of the wall clock running time would be desirable.

- In terms of the image experiments, I do expect to see results with larger datasets/models, though not absolutely necessary.

Typos:
- Page 5, Eq. 9, x_i should be h_i instead?
- Page 9, beta^l_j = 0 and ??^_j = 1

**Experience Assessment:**

I have read many papers in this area.

**Review Assessment: Checking Correctness Of Derivations And Theory:**

I assessed the sensibility of the derivations and theory.

**Review Assessment: Checking Correctness Of Experiments:**

I assessed the sensibility of the experiments.

**Review Assessment: Thoroughness In Paper Reading:**

I read the paper at least twice and used my best judgement in assessing the paper.

---

> ### Author Response · Authors · 2019-11-11
> **Author response to Reviewer 3**
>
> We appreciate Reviewer 3 for the carefully reading our work and providing valuable comments. We address your cons as follows:
>
>
> - Difference between PER and Huang et. al. 2018
> Yes, it is true that DBN (Huang et. al., 2018) also captures the interaction between hidden units though whitening. However, there are many cases PER and DBN have different behavior in making activations to follow the standard normal distribution since PER aims to match the distributions and DBN aims to whiten the activations. For instance, DBN cannot make change activations from a skewed distribution or a multimodal distribution having zero mean and the identity covariance matrix, unlike PER. This limitation of DBN can be found in Bilen & Vedaldi (2017) and Deecke et al. (2019) pointing out the inadequacy of normalizing multi-modal distributions by single mean and variance. To clarify this difference, we added a new figure (Fig. 1) and a new paragraph in P. 2-3 (last paragraph of the section 2) in the revised manuscript.
>
> -------
> Reference
>
> Lei Huang, Dawei Yang, Bo Lang, and Jia Deng. Decorrelated batch normalization. In IEEE Conference on Computer Vision and Pattern Recognition, 2018.
> Hakan Bilen and Andrea Vedaldi. Universal representations: The missing link between faces, text, planktons, and cat breeds. arXiv preprint arXiv:1701.07275, 2017.
> Lucas Deecke, Iain Murray, and Hakan Bilen. Mode normalization. In International Conference on Learning Representations, 2019.
>
>
> - Computational cost of PER
> Thanks for this great suggestion. As per Reviewer 3 pointed out, PER has non-negligible computational costs in backward pass having O(n d_l s) time complexity while BN has the time complexity of O(n d_l) where b is the size of mini-batch, s is the number of projection, and d_l is the number of hidden units in layer l. In terms of the wall clock running time, a vanilla network, BN, VCL, and PER take 0.071, 0.083, 0.087, and 0.093 seconds for a single forward/backward iteration in 11-layer CNN on a single NVIDIA TITAN X, respectively. The clarification and comparison of computational costs are added in section 4.3.1 of the revised manuscript.
>
>
> - Experiments on larger datasets/models
> We appreciate the suggestion from Reviewer 3. As Reviewer 3 suggested, we performed additional experiments on tiny ImageNet (a subset of ImageNet). It has 2x more training samples, 2x more categories, and 2x bigger image size. Besides, following the experiment given in VCL, we used 2x more filters in the experiment, i.e., 2x larger model. As other experiments performed in the original manuscript, we obtained better results than BN, VCL, and a vanilla network, and added the experiment in the revised manuscript.
>
>
> - Typos
> Thanks for carefully reviewing our manuscript. We modified the typos.

---

### Official Review · AnonReviewer1 · 2019-10-23
**Official Blind Review #1**

**Rating:** 6

**Review:**

This submission belongs to the general field of neural networks and sub-field of activation regularisation. In particular, this submission proposes a novel approach for activation regularisation whereby a distribution of activations within minibatch are regularised to have standard normal distribution. The approach, projected error function regularisation (PER), accomplishes that by minimising an upper-bound on 1-Wasserstein distance between empirical and standard normal distributions.

I think the idea described in this submission is interesting. Unfortunately, I have issues with 1) presentation, 2) experimental results, 3) English.

The PER is presented as an objective function that minimises an upperbound on 1-Wasserstein. I believe I have seen no evidence to the origin of PER other than it is the upper-bound on 1-Wasserstein. Therefore, I find it strange to see a presentation where first an objective function is introduced, then 1-Wasserstein is described, and after applying standard inequality you obtain an expression that is PER. The current presentation seems to indicate that before this derivation has been done no one new the connection between PER and the upper bound on 1-Wasserstein. I disagree and say that you obtained the upper bound on 1-Wasserstein and called it PER. For unknown reasons you decided to present first PER, then upper bound and finally claim connection. This is a mistake as it is not a connection but merely a consequence.

Simply looking up CIFAR-10 best numbers on any search engine I can find significantly better numbers. It is therefore unclear why did you decide to use sub-optimal configuration without commenting on that. The same applies to PTB and possibly to WikiText2.

There are numerous places where English is not adequate. For instance, "new perspective of concerning the target distribution".

Following the rebuttal stage where the authors have made significant changes to the manuscript I have decided to increase my assessment score.




**Experience Assessment:**

I have published in this field for several years.

**Review Assessment: Checking Correctness Of Derivations And Theory:**

I carefully checked the derivations and theory.

**Review Assessment: Checking Correctness Of Experiments:**

I carefully checked the experiments.

**Review Assessment: Thoroughness In Paper Reading:**

I read the paper thoroughly.

---

> ### Author Response · Authors · 2019-11-11
> **Author response to Reviewer 1**
>
> We appreciate Reviewer 1 for giving constructive feedback. Following your comments, we thoroughly revised the manuscript and believe the comments significantly improve the clarity of the manuscript. We address your three concerns as follows:
>
>
> - Presentation
> We sincerely thank Reviewer 1 for this insightful comment. As per Reviewer 1 pointed out, it is true that we obtained PER by applying the Minkowski inequality to 1-Wasserstein. In the original manuscript, we presented PER then derived PER from the 1-Wasserstein for emphasizing the difference between BN and PER, and now we admit that was a mistake. In the revised manuscript, we thoroughly revised the presentation as deriving the PER from the 1-Wasserstein and then explaining its difference with BN. We believe this change significantly improves the presentation of the manuscript and emphasizes the difference with existing methods even better.
>
> - Experimental result
> We thank for pointing out missing details in the experimental configuration. As Reviewer 1 indicated, the experimental configurations in the manuscript may be sub-optimal. However, to carefully compare PER with existing methods (BN, VCL, and L1 and L2 activation regularizations), we use the default hyperparameters of baseline models given in their papers. To clarify this point, we added explicit comments about this in each experiment and provided the benchmark results of literature in the result tables.
>
>
> - There are numerous places where English is not adequate.
> In response to Reviewer 1, we have very carefully proofread the manuscript again and corrected grammatical errors, typos, and inadequate expressions. We hope that Reviewer will notify us if there are still places where English is not adequate such as "new perspective of concerning the target distribution."

---

### Official Review · AnonReviewer2 · 2019-10-28
**Official Blind Review #2**

**Rating:** 6

**Review:**

This paper introduces "projected error function regularization loss" or PER, an alternative to batch normalization. PER is based on the Wasserstein metric. The experimental results show that PER outperforms batch normalization on CIFAR-10/100 with most activation functions. The authors also test their method on language modeling tasks.

Caveat: I'm not an expert in this domain. Hence, please take my rating with a large grain of salt.

Comments/questions:
- What's the computational cost of using PER over batch norm?
- Related to my other question: For the CIFAR-10 & CIFAR-100 comparison. What was the training time for BN vs PER?

**Experience Assessment:**

I do not know much about this area.

**Review Assessment: Checking Correctness Of Derivations And Theory:**

N/A

**Review Assessment: Checking Correctness Of Experiments:**

N/A

**Review Assessment: Thoroughness In Paper Reading:**

N/A

---

> ### Author Response · Authors · 2019-11-11
> **Author response to Reviewer 2**
>
> We thank Reviewer 2 for your time and efforts to point out the missing details of the original manuscript in computational cost that is an important issue when proposing a new regularizer. We added the benchmarking result with computational cost analysis in section 4.3.1 of the revised manuscript and address your comments as follows:
>
> - Computational cost of PER and BN
> In terms of time complexity, PER has the complexity of O(b d_l s) for projection operation where b is the size of mini-batch, s is the number of projection, and d_l is the number of hidden units in layer l. On the other hand, BN has O(b d_l) complexities for element-wise arithmetic operations and computations of mean and variance.
>
> - Training time for PER and BN in CIFAR
> In our wall clock running time measure, each training iteration takes 0.071 seconds for a vanilla network, 0.083 seconds for BN, 0.087 for VCL, and 0.093 seconds for PER on a single NVIDIA TITAN X.

---

### Author Response · Authors · 2019-11-11
**General statement**

We sincerely thank the reviewers for their insightful comments. We have uploaded a revised manuscript and summarize the major changes below.

1. In section 2, we added the paragraph illustrating the difference between PER, BN, and decorrelated BN.

2. In section 3, we changed the presentation of the proposed method as deriving the PER from the 1-Wasserstein and then explaining its difference with BN.

3. In section 4.1, we conducted and added an experiment on the larger dataset with the larger model.

4. In section 4.3.1, we added computational complexity analysis.

---

### Decision · Program_Chairs · 2019-12-19

**Decision:**

Accept (Poster)

**Comment:**

This paper presents an interesting and novel idea that is likely to be of interest to the community. The most negative reviewer did not acknowledge the author response. The AC recommends acceptance.